# Variational Inference in Mixed Probabilistic Submodular Models

**Josip Djolonga    Sebastian Tschiatschek    Andreas Krause**
Department of Computer Science, ETH Zürich
{josipd,tschiats,krausea}@inf.ethz.ch

## Abstract

We consider the problem of variational inference in probabilistic models with both log-submodular and log-supermodular higher-order potentials. These models can represent *arbitrary distributions* over binary variables, and thus generalize the commonly used pairwise Markov random fields and models with log-supermodular potentials only, for which efficient approximate inference algorithms are known. While inference in the considered models is #P-hard in general, we present efficient approximate algorithms exploiting recent advances in the field of discrete optimization. We demonstrate the effectiveness of our approach in a large set of experiments, where our model allows reasoning about preferences over sets of items with complements and substitutes.

## 1 Introduction

Probabilistic inference is one of the main building blocks for decision making under uncertainty. In general, however, this problem is notoriously hard even for deceptively simple-looking models and approximate inference techniques are necessary. There are essentially two large classes in which we can categorize approximate inference algorithms — those based on variational inference or on sampling. However, these methods typically do not scale well to large numbers of variables, or exhibit an exponential dependence on the model order, rendering them intractable for models with large factors, which can naturally arise in practice.

In this paper we focus on the problem of inference in point processes, i.e. distributions $P(A)$ over subsets $A \subseteq V$ of some finite ground set $V$. Equivalently, these models can represent arbitrary distributions over $|V|$ binary variables[1]. Specifically, we consider models that arise from submodular functions. Recently, Djolonga and Krause [1] discussed inference in *probabilistic submodular models (PSMs)*, those of the form $P(A) \propto \exp(\pm F(A))$, where $F$ is submodular. These models are called log-submodular (with the plus) and log-supermodular (with the minus) respectively. They generalize widely used models, e.g., pairwise purely attractive or repulsive Ising models and determinantal point processes (DPPs) [2]. Approximate inference in these models via variational techniques [1, 3] and sampling based methods [4, 5] has been investigated.

However, many real-world problems have neither purely log-submodular nor log-supermodular formulations, but can be naturally expressed in the form $P(A) \propto \exp(F(A) - G(A))$, where both $F(A)$ and $G(A)$ are submodular functions — we call these types of models *mixed PSMs*. For instance, in a probabilistic model for image segmentation there can be both attractive (log-supermodular) potentials, e.g., potentials modeling smoothness in the segmentation, and repulsive (log-submodular) potentials, e.g., potentials indicating that certain pixels should not be assigned to the same class. While the sampling based approaches for approximate inference are in general applicable to models

with both types of factors, fast mixing is only guaranteed for a subclass of all possible models and these methods may not scale well to large ground sets. In contrast, the variational inference techniques were only developed for either log-submodular or log-supermodular models.

In this paper we close this gap and develop variational inference techniques for mixed PSMs. Note that these models can represent arbitrary positive distributions over sets as any set function can be represented as the difference of a submodular and a supermodular function [6].[2] By exploiting recent advances in submodular optimization we formulate efficient algorithms for approximate inference that easily scale to large ground sets and enable the usage of large mixed factors.

**Applications/Models.** Mixed PSMs are natural models for a variety of applications — modeling of user preferences, 3D stereo reconstruction [7], and image segmentation [8, 9] to name a few. For instance, user preferences over items can be used for recommending products in an online marketing application and naturally capture the economic notions of *substitutes* and *complements*. Informally, item $a$ is a substitute for another item $b$ if, given item $b$, the utility of $a$ diminishes (log-submodular potentials); on the other hand, an item $c$ is a complement for item $d$ if, given item $d$, the utility of $c$ increases (log-supermodular potentials). Probabilistic models that can model substitutes of items are for example DPPs [2] and the facility location diversity (FLID) model [10]. In §4 we extend FLID to model both substitutes and complements which results in improved performance on a real-world product recommendation task. In terms of computer vision problems, non-submodular binary pairwise MRFs are widely used [8], e.g., as discussed above in image segmentation.

**Our contributions.** We generalize the variational inference procedure proposed in [1] to models containing both log-submodular and log-supermodular potentials, enabling inference in arbitrary distributions over binary variables. Furthermore, we provide efficient approximate algorithms for factor-wise coordinate descent updates enabling faster inference for certain types of models, in particular for rich scalable diversity models. In a large set of experiments we demonstrate the effectiveness of mixed higher-order models on a product recommendation task and illustrate the merit of the proposed variational inference scheme.

## 2   Background: Variational Inference in PSMs

**Submodularity.** Let $F\colon 2^V \to \mathbb{R}$ be a set function, i.e., a function mapping sets $A \subseteq V$ to real numbers. We will furthermore w.l.o.g assume that $V = \{1, 2, \ldots, n\}$. Formally, a function $F$ is called *submodular* if it satisfies the following *diminishing returns* property for all $A \subseteq B \subseteq V \setminus \{i\}$:

$$F(A \cup \{i\}) - F(A) \geq F(B \cup \{i\}) - F(B).$$

Informally, this property states that the gain of an item $i$ in the context of a smaller set $A$ is larger than its gain in the context of a larger set $B$. A function $G$ is called *supermodular* if $-G$ is submodular. A function $F$ is *modular*, if it is both submodular and supermodular. Modular functions $F$ can be written as $F(A) = \sum_{i \in A} m_i$ for some numbers $m_i \in \mathbb{R}$, and can be thus parameterized by vectors $\mathbf{m} \in \mathbb{R}^n$. As a shorthand we will frequently use $m(A) = \sum_{i \in A} m_i$.

**Probabilistic submodular models (PSMs).** PSMs are distributions over sets of the form

$$P(A) = \frac{1}{\mathcal{Z}} \exp(\pm F(A)),$$

where $\mathcal{Z} = \sum_{A \subseteq V} \exp(\pm F(A))$ ensures that $P(A)$ is normalized, and is often called the partition function. The distribution $P(A)$ is called *log-submodular* if the sign in the above definition is positive and *log-supermodular* if the sign is negative. These distributions generalize many well known classical models and have been effectively used for image segmentation [11], and for modeling diversity of item sets in recommender systems [10]. When $F(A) = m(A)$ is a modular function, $P(A) \propto \exp(F(A))$ is called *log-modular* and corresponds to a fully factorized distribution over $n$ *binary* random variables $X_1, \ldots, X_n$, where we have for each element $i \in V$ an associated variable $X_i$ indicating if this element is included in $A$ or not. The resulting distribution can be written as

$$P(A) = \frac{1}{\mathcal{Z}} \exp(m(A)) = \prod_{i \in A} \sigma(m_i) \prod_{i \notin A} \sigma(-m_i),$$

where $\sigma(u) = 1/(1 + e^{-u})$ is the sigmoid function.

**Variational inference and submodular polyhedra.** Djolonga and Krause [1] considered variational inference for PSMs, whose idea we will present here in a slightly generalized manner. Their approach starts by bounding $F(A)$ using functions of the form $m(A) + t$, where $m(A)$ is a modular function and $t \in \mathbb{R}$. Let us first analyze the *log-supermodular* case. If for all $A \subseteq V$ it holds that $m(A) + t \leq F(A)$, then we can bound the partition function $\mathcal{Z}$ as

$$\log \mathcal{Z} = \log \sum_{A \subseteq V} e^{-F(A)} \leq \log \sum_{A \subseteq V} e^{-m(A)-t} = \sum_{i=1}^{n} \log(1 + e^{-m_i}) - t.$$

Then, the idea is to optimize over the free parameters $\mathbf{m}$ and $t$ to find the best upper bound, or to equivalently solve the optimization problem

$$\min_{(\mathbf{m},t) \in \mathcal{L}(F)} \sum_{i=1}^{n} \log(1 + \exp(-m_i)) - t, \tag{1}$$

where $\mathcal{L}(F)$ is the set of all lower bounds of $F$, also known as the *generalized submodular lower polyhedron* [12]

$$\mathcal{L}(F) := \{(\mathbf{x}, t) \in \mathbb{R}^{n+1} \mid \forall A \subseteq V : x(A) + t \leq F(A)\}. \tag{2}$$

Djolonga and Krause [1] show that one obtains the same optimum if we restrict ourselves to $t = 0$ and one additional constraint, i.e., if we instead of $\mathcal{L}(F)$ use the base polytope $B(F)$ defined as

$$B(F) := \mathcal{L}(F) \cap \{(\mathbf{x}, 0) \in \mathbb{R}^{n+1} \mid x(V) = F(V)\}.$$

In words, it contains all modular lower bounds of $F$ that are *tight* at $V$ and $\emptyset$. Thanks to the celebrated result of Edmonds [13], one can optimize linear functions over $B(F)$ in time $O(n \log n)$. This, together with the fact that $\log(1 + e^{-u})$ is $\frac{1}{4}$-smooth, in turn renders the optimization problem (1) solvable via the Frank-Wolfe procedure [14, 15].

In the log-submodular case, we have to replace in problem (1) the minuses with pluses and use instead of $\mathcal{L}(F)$ the set of upper bounds. This set, denoted as $\mathcal{U}(F)$, defined by reversing the inequality sign in Equation (2), is called the *generalized submodular upper polyhedron* [12]. Unfortunately, in contrast to $\mathcal{L}(F)$, one can not easily optimize over $\mathcal{U}(F)$ and asking membership queries is an NP-hard problem. As discussed by Iyer and Bilmes [12] there are some special cases, like $M^{\natural}$-concave functions [16], where one can describe $\mathcal{U}(F)$, which we will discuss in § 3. Alternatively, which is the approach taken by [3], one can select a specific subfamily of $\mathcal{U}(F)$ and optimize over them.

## 3 Inference in Mixed PSMs

We consider mixed PSMs, i.e. probability distributions over sets that can be written in the form

$$P(A) \propto \exp(F(A) - G(A)),$$

where $F(A)$ and $G(A)$ are both submodular functions. Furthermore, we assume that $F$ and $G$ decompose as $F(A) = \sum_{i=1}^{m_F} F_i(A)$, and $G(A) = \sum_{i=1}^{m_G} G_i(A)$, where the functions $F_i$ and $G_i$ are all submodular. Note that this is not a limiting assumption, as submodular functions are closed under addition and we can always take $m_F = m_G = 1$, but such a decomposition will sometimes allow us to obtain better bounds. The corresponding distribution has the form

$$P(A) \propto \prod_{j=1}^{m_F} \exp(F_j(A)) \prod_{j=1}^{m_G} \exp(-G_j(A)). \tag{3}$$

Similarly to the approach by Djolonga and Krause [1], we perform variational inference by upper bounding $F(A) - G(A)$ by a modular function parameterized by $\mathbf{m}$ and a constant $t$ such that

$$F(A) - G(A) \leq \mathbf{m}(A) + t \text{ for all } A \subseteq V. \tag{4}$$

This upper bound induces the log-modular distribution $Q(A) \propto \exp(\mathbf{m}(A) + t)$. Ideally, we would like to select $(\mathbf{m}, t)$ such that the partition function of $Q(A)$ is as small as possible (and thus our approximation of the partition function of $P(A)$ is as tight as possible), i.e., we aim to solve

$$\min_{(\mathbf{m},t) \in \mathcal{U}(F-G)} t + \sum_{i=1}^{|V|} \log(1 + \exp(m_i)). \tag{5}$$

Optimization (and even membership checks) over $\mathcal{U}(F - G)$ is in general difficult, mainly because of the structure of $\mathcal{U}(F - G)$, which is given by $2^n$ inequalities. Thus, we seek to perform a series of *inner* approximations of $\mathcal{U}(F - G)$ that make the optimization more tractable.

**Approximating** $\mathcal{U}(F - G)$. In a first step we approximate $\mathcal{U}(F - G)$ as $\mathcal{U}(F) - \mathcal{L}(G) \subseteq \mathcal{U}(F - G)$, where the summation is understood as a Minkowski sum. Then, we can replace $\mathcal{L}(G)$ by $B(G)$ without losing any expressive power, as shown by the following lemma (see [3][Lemma 6]).

**Lemma 1.** *Optimizing problem* (5) *over* $\mathcal{U}(F) - \mathcal{L}(G)$ *and over* $\mathcal{U}(F) - B(G)$ *yields the same optimum value.*

This lemma will turn out to be helpful when we shortly describe our strategy for minimizing (5) over $\mathcal{U}(F) - B(G)$ as it will render some of our subproblems convex optimization problems over $B(G)$— these subproblems can then be efficiently solved using the Frank-Wolfe algorithm as proposed in [1] by noting that a greedy algorithm can be used to solve linear optimization problems over $B(G)$ [17].

By assumption, $F(A)$ and $G(A)$ are composed of simpler functions. First, because $G = \sum_{j=1}^{m_G} G_j$, it holds that $B(G) = \sum_{j=1}^{m_G} B(G_j)$ (see e.g. [18]). Second, even though it is hard to describe $\mathcal{U}(F)$, it might hold that $\mathcal{U}(F_i)$ has a tractable description, which leads to the natural inner approximation $\mathcal{U}(F) \supseteq \sum_{j=1}^{m_F} \mathcal{U}(F_j)$. To wrap up, we performed the following series of inner approximations

$$
\begin{array}{ccccc}
\mathcal{U}(F - G) & \supseteq & \mathcal{U}(F) & - & B(G) \\
& & \rotatebox{90}{$\subseteq$} & & \| \\
& \supseteq & \sum_{j=1}^{m_F} \mathcal{U}(F_j) & - & \sum_{j=1}^{m_G} B(G_j)
\end{array} ,
$$

which we then use to approximate $\mathcal{U}(F - G)$ in problem (5) before solving it.

**Optimization.** To solve the resulting problem we use a block coordinate descent procedure. Let us first rewrite the problem in a form that enables us to easily describe the algorithm. Let us write our resulting approximation as

$$
(\mathbf{m}, t) = \sum_{j=1}^{m_F} (\mathbf{f}_j, t_j) - \sum_{j=1}^{m_G} (\mathbf{g}_j, 0),
$$

where we have constrained $(\mathbf{f}_j, t_j) \in \mathcal{U}(F_j)$ and $\mathbf{g}_j \in B(G_j)$. The resulting problem is then to solve

$$
\min_{(\mathbf{f}_j, t_j) \in \mathcal{U}(F_j), \mathbf{g}_j \in B(G_j)} \underbrace{\sum_{j=1}^{m_F} t_j + \sum_{i=1}^{n} \log \left[ 1 + \exp(\sum_{j=1}^{m_F} f_{j,i} - \sum_{j=1}^{m_G} g_{j,i}) \right]}_{=: T((\mathbf{f}_j, t_j)_{j=1,\ldots,m_F}, (\mathbf{g}_j)_{j=1,\ldots,m_G})} . \tag{6}
$$

Then, until convergence, we pick one of the $m_G + m_F$ blocks uniformly at random and solve the resulting optimization problem, which we now show how to do.

**Log-supermodular blocks.** For a log-supermodular block $j$, minimizing (6) over $\mathbf{g}_j$ is a smooth convex optimization problem and we can either use the Frank-Wolfe procedure as in [1], or the divide-and-conquer algorithm (see e.g. [19]). In particular, if we use the Frank-Wolfe procedure we perform a block coordinate descent step with respect to (6) by iterating the following until we achieve some desired precision $\epsilon$: Given the current $\mathbf{g}_j$, we compute $\nabla_{\mathbf{g}_j} T$ and use the greedy algorithm to solve $\arg\min_{\mathbf{x} \in B(G_j)} \langle \mathbf{x}, \nabla_{\mathbf{g}_j} T \rangle$ in $O(n \log n)$ time. We then update $\mathbf{g}_j$ to $(1 - \frac{2}{k+2})\mathbf{x} + \frac{2}{k+2}\mathbf{g}_j$, where $k$ is the iteration number.

**Log-submodular blocks.** As we have already mentioned, this optimization step is much more challenging. One procedure, which is taken by [1], is to consider a set of $2^n$ points inside $\mathcal{U}(F_j)$ and optimize over them, which turns out to be a submodular minimization problem. However, for specific subfamilies, we can better describe $\mathcal{U}(F_j)$. One particularly interesting subfamily is that of $M^\natural$-concave functions [16], which have been studied in economics [20].

A set function $H$ is called $M^\natural$-concave if $\forall A, B \subseteq V, i \in A \setminus B$ it satisfies

$$
H(A) + H(B) \leq H(A \setminus \{i\}) + H(B \cup \{i\}) \text{ or}
$$
$$
\exists j \in B \setminus A : H(A) + H(B) \leq H((A \setminus \{i\}) \cup \{j\}) + H((B \cup \{i\}) \setminus \{j\}).
$$

Equivalently, these functions can be defined through the so called *gross substitutability* property known in economics. It turns out that $M^\natural$-concave set functions are also submodular. Examples of these functions include facility location functions, matroid rank functions, monotone concave over cardinality functions, etc. [16]. For example, $H(A) = \max_{i \in A} h_i$ for $h_i \geq 0$ is $M^\natural$-concave, which we will exploit in our models in the experimental section.

Returning to our discussion of optimizing (6), if $F_j$ is an $M^\natural$-concave function, we can minimize (6) over $(\mathbf{f}_j, t_j) \in \mathcal{U}(F_j)$ to arbitrary precision in polynomial time. Therefore, we can, similarly as in [1], use the Frank-Wolfe algorithm by noting that a polynomial time algorithm for computing $\arg\min_{\mathbf{x} \in \mathcal{U}(F_j)} \langle \mathbf{x}, \nabla_{(\mathbf{f}_j, t_j)} T \rangle$ exists [20]. Although the minimization can be performed in polynomial time, it is a very involved algorithm. We therefore consider an inner approximation $\check{\mathcal{U}}(F_j) := \{(\mathbf{m}, 0) \in \mathbb{R}^{n+1} \mid \forall A \subseteq V : F(A) \leq \mathbf{m}(A)\} \subseteq \mathcal{U}(F_j)$ of $\mathcal{U}(F_j)$ over which we can more efficiently approximately minimize (6). As pointed out by Iyer and Bilmes [12], for $M^\natural$ functions $F_j$ the polyhedron $\check{\mathcal{U}}(F_j)$ can be characterized by $\mathcal{O}(n^2)$ inequalities as follows:

$$\check{\mathcal{U}}(F_j) := \cup_{A \subseteq V} \{(\mathbf{m}, 0) \in \mathbb{R}^{n+1} \mid \forall i \in A : m_i \leq F_j(A) - F_j(A \setminus \{i\}),$$
$$\forall k \notin A : m_j \geq F_j(A \cup \{k\}) - F_j(A),$$
$$\forall i \in A, k \notin A : m_i - m_k \leq F_j(A) - F_j((X \cup \{i\}) \setminus \{k\})\}.$$

We propose to use Algorithm 1 for minimizing over $\check{\mathcal{U}}(F_j)$. Given a set $A$ where we want our modular approximation to be exact at, the algorithm iteratively minimizes the partition function of a modular upper bound on $F_j$. Clearly, after the first iteration of the algorithm $(\mathbf{m}, 0)$ is an upper bound on $F_j$. Furthermore, the partition function corresponding to that bound decreases monotonically over the iterations of the algorithm. Several heuristics can be used to select $A$—in the experiments we determined $A$ as follows: We initialized $B = \emptyset$ and then, while $0 < \max_{i \in V \setminus A} F(B \cup \{i\}) - F(B)$, added $i$ to $B$, i.e. $B \cup \{\arg\max_{i \in V \setminus B} F(B \cup \{i\}) - F(B)\}$. We used the final $B$ of this iteration as our tight set $A$.

---

**Algorithm 1** Modular upper bound for $M^\natural$-concave functions

---

**Require:** $M^\natural$ function $F$, tight set $A$ s.t. $\mathbf{m}(A) = F(A)$ for the returned $\mathbf{m}$
   Initialize $\mathbf{m}$ randomly
   **for** $l = 1, 2, \ldots$, max. nr. of iterations **do**         ▷ Alt. minimize $\mathbf{m}$ over coeff. corresponding to $A$ and $V \setminus A$
      $\forall i \in A : m_i = \min\{F(A) - F(A \setminus \{i\}), \min_{k \in V \setminus A} m_k + F(A) - F((A \cup \{i\}) \setminus \{k\})\}$
      $\forall k \notin A : m_k = \max\{F(A \cup \{k\}) - F(A), \max_{i \in A} m_i - F(A) + F((A \cup \{i\}) \setminus \{k\})\}$
   **end for**
   **return** Modular upper bound $\mathbf{m}$ on $F$

---

## 4 Examples of Mixed PSMs for Modelling Substitutes and Complements

In our experiments we consider probabilistic models that take the following form:

$$H(A; \alpha, \beta) = \sum_{i \in A} u_i + \alpha \sum_{l=1}^{L} \underbrace{\left( \max_{i \in A} r_{l,i} - \sum_{i \in A} r_{l,i} \right)}_{F_l(A)} - \beta \sum_{k=1}^{K} \underbrace{\left( \max_{i \in A} a_{k,i} - \sum_{i \in A} a_{k,i} \right)}_{G_k(A)}, \qquad (7)$$

where $\alpha, \beta \in \{0, 1\}$ switch on/off the repulsive and attractive capabilities of the model, respectively. We would like to point out that even though $\sum_{l=1}^{L} F_l(A)$ is not $M^\natural$-concave, each summand $F_l$ is, which we will exploit in the next section. The model is parameterized by the vector $\mathbf{u} \in \mathbb{R}^{|V|}$, and the weights $(\mathbf{r}_l)_{l \in [L]}, \mathbf{r}_l \in \mathbb{R}_{\geq 0}^{|V|}$ and $(\mathbf{a}_k)_{k \in [K]}, \mathbf{a}_k \in \mathbb{R}_{\geq 0}^{|V|}$, which will be explained shortly. From the general model (7) we instantiate four different models as explained in the following.

*Log-modular model.* The log-modular model $P_{\text{mod}}(A)$ is instantiated from (7) by setting $\alpha = \beta = 0$, i.e. $F_{\text{mod}}(A) := H(A; 0, 0)$ and serves as a baseline model. This model cannot capture any dependencies between items and corresponds to a fully factorized distribution over the items in $V$.

*Facility location diversity model (FLID).* This model is instantiated from (7) by setting $\alpha = 1, \beta = 0$, i.e. $F_{\text{FLID}}(A) := H(A; 1, 0)$, and is known as facility location diversity model (FLID) [10]. Note that

this induces a log-submodular distribution. The FLID model parameterizes all items $i$ by an item quality $u_i$ and an $L$-dimensional vector $\mathbf{r}_{\cdot,i} \in \mathbb{R}^L_{\geq 0}$ of latent properties. The model assigns a negative penalty $F_l(A) = \max_{i \in A} r_{l,i} - \sum_{i \in A} r_{l,i}$ whenever at least two items in $A$ have the same latent property (the corresponding dimensions of $\mathbf{r}_l$ are $> 0$) — thus the model explicitly captures repulsive dependencies between items.[3] Speaking in economic terms, items with similar latent representations can be considered as *substitutes* for each other. The FLID model has been shown to perform on par with DPPs on product recommendation tasks [10].

*Facility location complements model (FLIC).* This model is instantiated from (7) by setting $\alpha = 0, \beta = 1$, i.e. $F_{\mathrm{FLIC}}(A) := H(A; 0, 1)$ and defines a log-supermodular probability distribution. Similar to FLID, the model parameterizes all items $i$ by an item quality $u_i$ and a $K$-dimensional vector $\mathbf{a}_{\cdot,i} \in \mathbb{R}^K_{\geq 0}$ of latent properties. In particular, there is a gain of $G_k(A) = \sum_{i \in A} a_{k,i} - \max_{i \in A} a_{k,i}$ if at least two items in $A$ have the same property $k$ (i.e. for both items the corresponding dimensions of $\mathbf{a}_k$ are $> 0$). In this way, FLIC captures attractive dependencies among items and assigns high probabilities to sets of items that have similar latent representations — items with similar latent representations would be considered as *complements* in economics.

*Facility location diversity and complements model (FLDC).* This model is instantiated from (7) via $F_{\mathrm{FLDC}}(A) := H(A; 1, 1)$. Hence it combines the modelling power of the log-submodular and log-supermodular models and can explicitly represent attractive and repulsive dependencies. In this way, FLDC can represent *complements and subsitutes* for the items in $V$. The induced probability distribution is neither log-submodular nor log-supermodular.

## 5 Experiments

### 5.1 Experimental Setup

**Dataset.** We use the Amazon baby registry dataset [21] for evaluating our proposed variational inference scheme. This dataset is a standard dataset for benchmarking diversity models and consists of baby registries collected from Amazon. These registries are split into sub-registries according to 13 different product categories, e.g. *safety* and *carseats*. Every category contains 32 to 100 different items and there are $\approx 5.000$ to $\approx 13.300$ sub-registries per category.

**Product recommendation task.** We construct a realistic product recommendation task from the registries of every category as follows. Let $\mathcal{D} = (S_1, \ldots, S_n)$ denote the registries from one category. From this data, we create a new dataset

$$\hat{\mathcal{D}} = \{(S \setminus \{i\}, i) \mid S \in \mathcal{D}, |S| \geq 2, i \in S\}, \tag{8}$$

i.e., $\hat{\mathcal{D}}$ consists of tuples, where the first element is a registry from $\mathcal{D}$ with one item removed, and the second element is the removed item. The product recommendation task is to predict $i$ given $S \setminus \{i\}$. For evaluating the performance of different models on this task we use the following two metrics: *accuracy* and *mean reciprocal rank*. Let us denote the recommendations of a model given a partial basket $A$ by $\sigma_A : V \to [n]$, where $\sigma_A(a) = 1$ means that product $a$ is recommended highest, $\sigma_A(b) = 2$ means that product $b$ is recommended second highest, etc. Then, the accuracy is computed as $\mathrm{Acc} = \frac{1}{|\hat{\mathcal{D}}|} \sum_{(S',i) \in \hat{\mathcal{D}}} [i = \sigma_{S'}^{-1}(1)]$. The mean reciprocal rank (MRR) is defined as $\mathrm{MRR} = \frac{1}{|\hat{\mathcal{D}}|} \sum_{(S',i) \in \hat{\mathcal{D}}} \frac{1}{\sigma_{S'}^{-1}(i)}$. For our models we consider predictions according to the posterior probability of the model given a partial basket $A$ under the constraint that exactly a single item is to be added, i.e. $\sigma_A(i) = k$ if product $i$ achieves the $k$-th largest value of $P(j|A) = \frac{P(\{j\} \cup A)}{\sum_{j' \in V \setminus A} P(\{j'\} \cup A)}$ for $j \in V \setminus A$ (ties are broken arbitrarily).

### 5.2 Mixed Models for Product Recommendation

We learned the models described in the previous section using the training data of the different categories. In case of the modular model, the parameters $\mathbf{u}$ were set according to the item frequencies in the training data. FLID, FLIC and FLDC were learned using noise contrastive estimation (NCE) [22, 10]. We used stochastic gradient descent for optimizing the NCE objective, created 200.000 noise

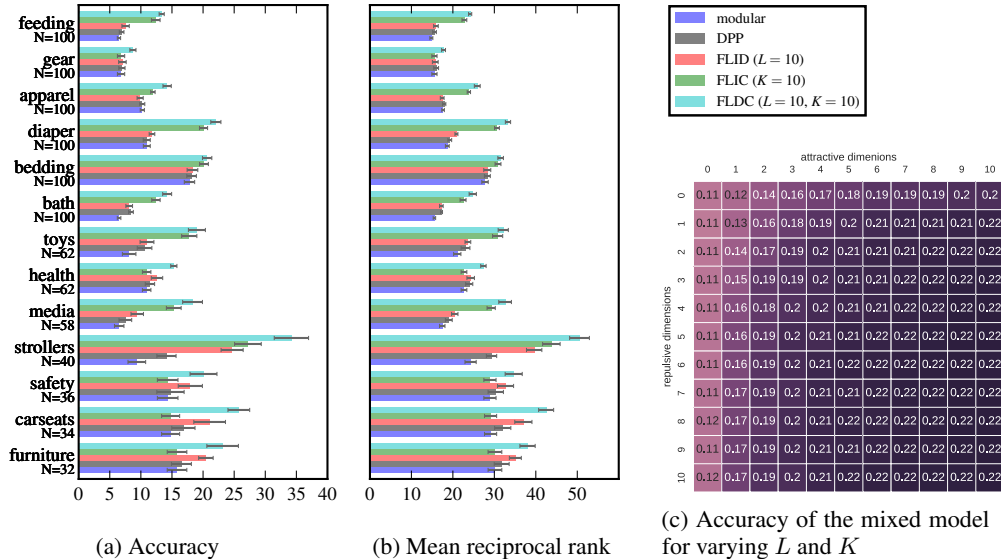

| | attractive dimensions | | | | | | | | | | |
|---|---|---|---|---|---|---|---|---|---|---|---|
| | 0 | 1 | 2 | 3 | 4 | 5 | 6 | 7 | 8 | 9 | 10 |
| 0 | 0.11 | 0.12 | 0.14 | 0.16 | 0.17 | 0.18 | 0.19 | 0.19 | 0.19 | 0.2 | 0.2 |
| 1 | 0.11 | 0.13 | 0.16 | 0.18 | 0.19 | 0.2 | 0.21 | 0.21 | 0.21 | 0.21 | 0.22 |
| 2 | 0.11 | 0.14 | 0.17 | 0.19 | 0.2 | 0.21 | 0.21 | 0.21 | 0.22 | 0.22 | 0.22 |
| 3 | 0.11 | 0.15 | 0.19 | 0.19 | 0.2 | 0.21 | 0.21 | 0.22 | 0.22 | 0.22 | 0.22 |
| 4 | 0.11 | 0.16 | 0.18 | 0.2 | 0.2 | 0.21 | 0.21 | 0.22 | 0.22 | 0.22 | 0.22 |
| 5 | 0.11 | 0.16 | 0.19 | 0.2 | 0.21 | 0.21 | 0.22 | 0.22 | 0.22 | 0.22 | 0.22 |
| 6 | 0.11 | 0.16 | 0.19 | 0.2 | 0.21 | 0.22 | 0.22 | 0.22 | 0.22 | 0.22 | 0.22 |
| 7 | 0.11 | 0.17 | 0.19 | 0.2 | 0.21 | 0.21 | 0.22 | 0.22 | 0.22 | 0.22 | 0.22 |
| 8 | 0.12 | 0.17 | 0.19 | 0.2 | 0.21 | 0.22 | 0.22 | 0.22 | 0.22 | 0.22 | 0.22 |
| 9 | 0.11 | 0.17 | 0.19 | 0.2 | 0.21 | 0.22 | 0.22 | 0.22 | 0.22 | 0.22 | 0.22 |
| 10 | 0.12 | 0.17 | 0.19 | 0.2 | 0.21 | 0.22 | 0.22 | 0.22 | 0.22 | 0.22 | 0.22 |

(a) Accuracy  (b) Mean reciprocal rank  (c) Accuracy of the mixed model for varying $L$ and $K$

Figure 1: *(a,b)* Accuracy and MRR on the product recommendation task. For all datasets, the mixed FLDC model has the best performance. For datasets with small ground set (furniture, carseats, safety) FLID performs better than FLIC. For most other datasets, FLIC outperforms FLID. *(c)* Accuracy of FLDC for different numbers of latent dimensions $L$ and $K$ on the diaper dataset. FLDC ($L, K > 0$) performs better than FLID ($K = 0$) and FLIC ($L = 0$) for the same value of $L + K$.

samples from the modular model and made 100 passes through the data and noise samples. We then used the trained models for the product recommendation task from the previous section and estimated the performance metrics using 10-fold cross-validation. We used $K = 10$, $L = 10$ dimensions for the weights (if applicable for the corresponding model). The results are shown in Figure 1. For reference, we also report the performance of DPPs trained with EM [21]. Note that for all categories the mixed FLDC models achieve the best performance, followed by FLIC and FLID. For categories with more than 40 items (with the exception of *health*), FLIC performs better than FLID. The modular model performs worst in all cases. As already observed in the literature, the performance of FLID is similar to that of DPPs [10]. For categories with small ground sets (*safety, furniture, carseats*), there is no advantage of using the higher-order attractive potentials but the repulsive higher-order potentials improve performance significantly. However, in combination with repulsive potentials the attractive potentials enable the model to improve performance over models with only repulsive higher-order potentials.

### 5.3   Impact of the Dimension Assignment

In Figure 1c we show the accuracy of FLDC for different numbers of latent dimensions $L$ and $K$ for the category *diaper*, averaged over the 10 cross-validation folds. Similar results can be observed for the other categories (not shown here because of space constraints). We note that the best performance is achieved only for models that have both repulsive and attractive components (i.e. $L, K > 0$). For instance, if one is constrained to use only 10 latent dimensions in total, i.e. $L + K = 10$, the best performance is achieved for the settings $L = 3, K = 7$ and $L = 2, K = 8$.

### 5.4   Quality of the Marginals

In this section we analyze the quality of the marginals obtained by the algorithm proposed in Section 3. Therefore we repeat the following experiment for all baskets $S$, $|S| \geq 2$ in the the held out test data. We randomly select a subset $S' \subset S$ of 1 to $|S| - 1$ items and a subset $S'' \subset V \setminus S$ with $|S''| = \lfloor |V \setminus S|/2 \rfloor$, of items not present in the basket. Then we condition our distribution on the event that the items in $S'$ are present and the items $S''$ are not present i.e. we consider the distribution $P(A \mid S' \subseteq A, S'' \cap A = \emptyset)$. This conditioning is supposed to resemble a fictitious product recommendation task in which we condition on items already selected by a user and exclude items which are of no interest to the user (for instance, according to the user's preferences). We then compute a modular approximation to the posterior distribution using the algorithm from Section 3,

Table 1: AUC for the considered models on the product recommendation task based on the posterior marginals. The best result for every dataset is printed in bold. For datasets with at most 62 items, FLDC has the highest AUC, while for larger datasets FLIC and FLDC have similar AUC. This indicates a good quality of the marginals computed by the proposed approximate inference procedure.

| Dataset | Modular | FLID | FLIC | FLDC |
|---------|---------|------|------|------|
| safety | 0.731304 | 0.756981 | 0.731269 | **0.761168** |
| furniture | 0.701840 | 0.739646 | 0.702100 | **0.759979** |
| carseats | 0.717463 | 0.770085 | 0.735472 | **0.781642** |
| strollers | 0.727055 | 0.794655 | 0.827800 | **0.849767** |
| health | 0.750271 | 0.754185 | 0.756873 | **0.758586** |
| bath | 0.692423 | 0.705051 | 0.730443 | **0.732407** |
| media | 0.666509 | 0.667848 | 0.758552 | **0.780634** |
| toys | 0.724763 | 0.729089 | 0.765474 | **0.777729** |
| bedding | 0.741786 | 0.744443 | **0.771159** | 0.764595 |
| apparel | 0.700694 | 0.696010 | 0.778067 | **0.779665** |
| diaper | 0.685051 | 0.700543 | **0.787457** | 0.787274 |
| gear | 0.687686 | 0.688116 | 0.687501 | **0.688885** |
| feeding | 0.686240 | 0.686845 | **0.744043** | 0.739921 |
| *Average* | 0.708698 | 0.725653 | 0.752016 | **0.766327** |

and recommend items according to these approximate marginals. For evaluation, we compute the AUC for the product recommendation task and average over the test set data. We found that for different models different modular upper/lower bounds gave the best results. In particular, for FLID we used the upper bound given by Algorithm 1 to bound each summand $F_l(A)$ in the facility location term separately. For FLIC and FLID we optimized the lower bound on the partition function by lower-bounding $\sum_{l=1}^{L} F_l(A)$ and upper bounding $\sum_{k=1}^{K} G_k(A)$, as suggested in [1]. For approximate inference in FLIC and FLDC we did not *split* the facility location terms and bounded them as a whole. The results are summarized in Table 1. We observe that FLDC has the highest AUC for all datasets with at most 62 items. For larger datasets, FLDC and FLIC have roughly the same performance and are superior to FLID and the modular model. These findings are similar to those from the previous section and confirm a good quality of the marginals computed from FLDC and FLIC by the proposed approximate inference procedure.

## 6  Related Work

Variational inference in general probabilistc log-submodular models has been first studied in [1]. The authors propose L-FIELD, an approach for approximate inference in both log-submodular and log-supermodular models based on super- and sub-differentials of submodular functions. In [3] they extended their work by relating L-FIELD to the minimum norm problem for submodular minimization, rendering better scalable algorithms applicable to variational inference in log-submodular models. The forementioned works can only be applied to models that contain either log-submodular or log-supermodular potentials and hence do not cover the models considered in this paper.

While the MAP solution in mixed models is known to be NP-hard, there are approximate methods for its computation based on iterative majorization-minimization (or minorization-maximization) procedures [23, 24]. In [9] the authors consider mixed models in which the supermodular component is restricted to a tree-structured cut, and provide several algorithms for approximate MAP computation. In contrast to our work, these methods are non-probabilistic and only provide an approximate MAP solution without any notion of uncertainty.

## 7  Conclusion

We proposed efficient algorithms for approximate inference in mixed submodular models based on inner approximations of the set of modular bounds on the corresponding energy functions. For many higher-order potentials, optimizing a modular bound over this inner approximation is tractable. As a consequence, the approximate inference problem can be approached by a block coordinate descent procedure, tightening a modular upper bound over the individual higher-order potentials in an iterative manner. Our approximate inference algorithms enable the computation of approximate marginals and can easily scale to large ground sets. In a large set of experiments, we demonstrated the effectiveness of our approach.

**Acknowledgements.** The authors acknowledge fruitful discussions with Diego Ballesteros. This research was supported in part by SNSF grant CRSII2-147633, ERC StG 307036, a Microsoft Research Faculty Fellowship and a Google European Doctoral Fellowship.

## Footnotes

[1]Distributions over sets $A \subseteq V$ are isomorphic to distributions over $|V|$ binary variables, where each binary variable corresponds to an indicator whether a certain element is included in $A$ or not.

[2] As the authors in [6] note, such a decomposition can be in general hard to find.

[3]Clearly, also attractive dependencies between items can thereby be modeled implicitly.

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
