[Reviews · NeurIPS 2016]

Reviewer 1

Summary

The paper studies the problem of performing variational inference in probabilistic models with both log-submodular and log-supermodular potentials. In a log-submodular (resp. log-supermodular) model, the probability of a set A is proportional to exp(F(A)) (resp. exp(-F(A))), where F is a submodular function. Both log-submodular and log-supermodular models have been studied and inference algorithms have been proposed for each of them individually. The current paper studies the more general setting where the probability of a set A is proportional to exp(F(A) - G(A)), where F and G are submodular. This is a very general setting, since any set function can be expressed as the difference of two submodular functions. By combining insights from previous work, the paper gives an approach for performing variational inference for these models under assumptions on F, and in particular that F belongs to the class of M^#-concave functions for which the recent work [12] has established certain useful properties.

Qualitative Assessment

The setting considered is very general and thus from a theoretical point of view it is very difficult to provide a principled approach. In order to make progress, the paper makes certain assumptions on the functions, and in particular it considers the case when the function F is M^#-concave. The authors motivate this assumption by providing supporting examples and some experiments showing that the proposed approach is beneficial. On a technical level, the variational inference approach does not seem provide any novel conceptual contributions, and the main technical ideas come from the previous work. The experiments show that the approach may have some impact, but the conceptual contributions do not seem to be very clear. The clarity of the presentation could be improved. In particular, in the main Section 3 describing the variational approach, it may be helpful to have explicit theorem statement stating all the assumptions made upfront, instead of gradually revealing assumptions/approximations in the text. == post-rebuttal answer== I have read the entire rebuttal.

Confidence in this Review

2-Confident (read it all; understood it all reasonably well)


Reviewer 2

Summary

In this work, the authors extended the work of Djolonga and Krause to the case of mixed log-submodular and log-supermodular probability distributions. The paper is well written, gives proper references and presents an innovative idea. Overall, I appreciate the paper and would like to discuss with the authors. The approach is quite straightforward and might have a limited impact, but I think it is worth publishing it at NIPS as this extends considerably the types of models that can be handled using Djolonga and Krause idea.

Qualitative Assessment

The introduction and explanation of Variational Inference in PSM is very good. I did not know about it and could understand the whole paper easily. The work might be seen as incremental: the idea of bounding F(A) - G(A) by m(A) + t is exactly the same as Djolonga and Krause. The technical details are well explained and might be enough for a publication at NIPS. In the experiments, the explanation about why optimizing the NCE is submodular would have been welcome. The evaluation has been made directly on the final task, I'm not sure how much the method is an approximation of the exact marginals. I would have been welcome to understand in which case the approximation due to the bound is much worse than the exact evaluation of the partition function or a sampled approximation of it (e.g. using importance sampling or MCMC).

Confidence in this Review

1-Less confident (might not have understood significant parts)


Reviewer 3

Summary

In this paper, the authors develop a variational inference framework for discrete probabilistic models, where the density is given as the exponent of the difference of two submodular functions. They provide a theoretical guarantee for their formulation and develop an optimization procedure based on a block coordinate descent. They empirically evaluate the proposed algorithms with one real-world dataset (popular benchmark data for DPPs).

Qualitative Assessment

I think this paper is an interesting extension of probabilistic submodular models. The model can express any discrete density, which, of course, means that either of existing important log-submodular or log-supermodular model such as determinantal point processes (DPPs) and binary Markov random fields are also included. Although each element for the development of the framework is not necessarily new (or it looks a collection of existing results), this paper would be the first one that discusses a variational inference for the mixed model. By the way, since the optimization problem (6) is still a computationally difficult one, I wonder how good a solution by the block coordinate descent approach empirically is. Or, how sensitive a solution is depending on the choice of initials etc. These questions could be clear somehow by performing empirical investigation (with, maybe, synthetic data) although, unfortunately, the current paper does not include such results.

Confidence in this Review

2-Confident (read it all; understood it all reasonably well)


Reviewer 4

Summary

The authors introduce approximate methods to estimate the log partition function and marginals of graphical models which have both log-submodular and log-supermodular high-order potentials, using variational methods over the generalized submodular lower and upper polyhedra. Experiments on real world data demonstrate the value of the approach.

Qualitative Assessment

The paper provides helpful background and motivation, and is mostly well written. I am not well versed in the earlier work mentioned. In Sec 3, please could you clarify which parts are novel contributions beyond earlier work including [1]? Lines 135-136: given the differences in difficulty of updating the m_G and m_F blocks, can you comment on typical improvements obtained, so could it make sense to choose updates not uniformly at random (but instead emphasizing the updates that might be estimated to yield more improvement for time spent, if possible, or similar)? Can you comment on when we should expect the approximations to perform well/badly? Would it be helpful to try small models where exact inference is possible to explore this? The experiments on large real data sets look convincing to demonstrate improvement in important settings of FLDC over DPP and FLID. Are those state of the art comparisons? Sec 6 mentions settings where approximate MAP inference is feasible. Have perturb-and-MAP ideas been explored as alternative approaches for marginal inference?

Confidence in this Review

1-Less confident (might not have understood significant parts)


Reviewer 5

Summary

This paper considers variational inference in probabilistic models involving log-submodular and log-supermodular potentials. Though inference in this mixed model is hard, they provide a heuristic method for this task, under the assumption that the submodular functions further satisfy M♮ convexity. Numerical experiment for recommendation system is presented and their algorithm exhibits a reasonable quality.

Qualitative Assessment

Although this paper is nicely written, I think the presented method lacks theoretical guarantee and justification in several points: - In the abstract, they claim that this paper presents the first efficient method for models with both log-submodular and log-supermodular potentials. However, their algorithm works only if the log-submodular parts admit M♮ concavity. Of course we can substitute another heuristic for general case, I do not think the title is appropriate. - The outer framework of their method is block coordinate descent. It was somewhat disappointing that they do not provide any theoretical guarantee on their method. For example, can we prove any theoretical guarantee provided that the exact upper bound for the M♮ concave parts is available? - They provide a faster algorithm for finding an upper bound of M♮ concave in Algorithm 1. Is this an exact algorithm or another heuristic? Overall, the main result is interesting but lacks theoretical guarantee or sufficient explanations. Minor comment: - Notation for the set of upper bounds of a set function F is not consistent: sometimes subscript U_F and sometimes U(F). - "M♮ concave" ("natural") is written in "M♯ concave" ("sharp") throughout the paper.

Confidence in this Review

2-Confident (read it all; understood it all reasonably well)


Reviewer 6

Summary

The authors consider the problem of inference in mixed submodular models, where the probability of a set "A" is proportional to exp(F(A) + G(A)), where F and G are submodular (i.e. -G is supermodular). Previous work was only able to handle a single submodular or supermodular function in the exponent. They then proceed to upper bound the partition function of the model by using a modular function that upper bounds F(A) - G(A). It is easy to calculate the partition function of a modular function. They then optimize over upper bounds to find an estimate of the partition function that is reasonably tight. Their optimization procedure involves an inner bound on the set of possible modular functions, then a block coordinate descent procedure using the Frank-Wolfe algorithm. The authors then empirically show that their approach achieves reasonable results in a product recommendation experiment.

Qualitative Assessment

The authors consider a useful generalization to inference in submodular models, which has been receiving a good amount of attention in recent conferences. Their paper is well written and synthesizes many results in the literature and can provide an example of how to use previously developed tools. However, the approach they take is not entirely novel compared to past work. With limited new theoretical results, to make the work a solid contribution it should have a more in-depth experimental analysis. Their experiment does not sufficiently compare to approaches outside the variational inference for submodular model framework. Specifically, they should compare to other approaches for estimating marginals, such as MCMC, and other approaches for estimation (e.g. logistic regression, SVM, boosting etc.) or mention why other approaches would not work. == post-rebuttal answer== Thank you for your rebuttal. After reading the other reviews I would agree that this is somewhat incremental, with many of the core ideas coming from previous work. However I found the problem to be interesting and believe that there are some nice algorithmic and modeling contributions. I would have liked to see a more extensive experimentation section with details on runtimes and other accuracy baselines to better judge the impact of this approach, especially since the algorithms are more heuristical in nature.

Confidence in this Review

2-Confident (read it all; understood it all reasonably well)